# Impact of Comorbid Polycystic Ovary Syndrome on Clinical and Laboratory Parameters in Female Adolescents with Metabolic Dysfunction-Associated Steatotic Liver Disease: A Cross-Sectional Study

**DOI:** 10.3390/jcm13195885

**Published:** 2024-10-02

**Authors:** Murat Keskin, Hanife Aysegul Arsoy, Ozlem Kara, Emre Sarandol, Nizameddin Koca, Yusuf Yilmaz

**Affiliations:** 1Department of Gastroenterology, Faculty of Medicine, KTO Karatay University, Konya 42020, Türkiye; keskinmd@hotmail.com; 2Department of Pediatric Gastroenterology, Bursa Yuksek Ihtisas Training and Research Hospital, University of Health Sciences, Bursa 16350, Türkiye; draysegulgastro@gmail.com; 3Department of Pediatric Endocrinology, Bursa Yuksek Ihtisas Training and Research Hospital, University of Health Sciences, Bursa 16350, Türkiye; dr.ozlemkara@hotmail.com; 4Department of Biochemistry, Faculty of Medicine, Bursa Uludağ University, Bursa 16059, Türkiye; sarandol@uludag.edu.tr; 5Department of Internal Medicine, Bursa Faculty of Medicine, University of Health Sciences, Bursa 16350, Türkiye; nizameddin.koca@sbu.edu.tr; 6Department of Gastroenterology, School of Medicine, Recep Tayyip Erdoğan University, Rize 53100, Türkiye; 7The Global NASH Council, Washington, DC 20037, USA

**Keywords:** metabolic dysfunction-associated steatotic liver disease, polycystic ovary syndrome, female adolescents, transient elastography, comorbidity, acanthosis nigricans

## Abstract

**Background**: Metabolic dysfunction-associated steatotic liver disease (MASLD) and polycystic ovary syndrome (PCOS) share several pathophysiological mechanisms. While the prevalence of MASLD has been extensively studied in PCOS populations, the occurrence of PCOS among female adolescents with transient elastography (TE)-confirmed MASLD in pediatric hepatology settings remains poorly characterized. This cross-sectional study aims to address this knowledge gap and elucidate potential clinical and biochemical differences between female adolescents with MASLD and comorbid PCOS compared to those without PCOS. **Methods**: The study cohort included 45 female adolescents with TE-diagnosed MASLD. Comparative analyses of clinical and laboratory parameters were performed between those with (n = 19) and those without (n = 26) comorbid PCOS, diagnosed according to the Rotterdam criteria. **Results**: Adolescents with MASLD and comorbid PCOS exhibited significantly higher weight, lower height, and increased waist circumference compared to those without PCOS. Additionally, the prevalence of acanthosis nigricans was significantly higher in the PCOS group (68.4% versus 34.6%, *p* = 0.025). Regarding laboratory parameters, serum phosphorus levels and liver enzymes—including aspartate aminotransferase, alanine aminotransferase, and gamma-glutamyl transferase—were significantly lower in adolescents with comorbid PCOS. However, no significant differences were observed in lipid profiles, glucose metabolism, or novel non-invasive biomarkers of MASLD. **Conclusions**: This study reveals distinct clinical and biochemical profiles in female adolescents with MASLD and comorbid PCOS compared to those without PCOS. These findings have the potential to inform and refine future screening protocols and diagnostic algorithms for these interrelated conditions, specifically tailored to pediatric hepatology settings.

## 1. Introduction

Polycystic ovary syndrome (PCOS) is closely linked to insulin resistance and an increased risk of type 2 diabetes, particularly among adolescents. Insulin resistance in PCOS intensifies hyperandrogenism, creating a cycle of metabolic and reproductive disturbances. Numerous studies, including those by Purwar et al. [1] and Witchel et al. [2], have shown that adolescents with PCOS often exhibit early signs of insulin resistance, increasing their susceptibility to metabolic syndrome and diabetes. These findings highlight the critical need for monitoring glucose metabolism and insulin sensitivity in young females diagnosed with PCOS.

The global prevalence of metabolic dysfunction-associated steatotic liver disease (MASLD) in pediatric populations has surged dramatically [3,4], closely mirroring the increasing rates of childhood obesity [5]. Recent estimates suggest that MASLD affects approximately 10% of all children and up to 35% of those with obesity [1]. The primary risk factors for pediatric MASLD include sedentary lifestyles, excessive caloric intake, poor dietary choices, and genetic susceptibility [6,7]. In female children and adolescents, MASLD may also be associated with other complex endocrine and metabolic disorders, notably PCOS, due to shared underlying pathophysiological mechanisms, including insulin resistance and hyperandrogenism [8]. While the association between MASLD and PCOS in adult women has been extensively investigated and subjected to numerous meta-analyses [9,10,11,12,13], research in adolescent populations remains comparatively limited but is garnering increasing attention.

The association between MASLD and PCOS in adolescents was first reported by Michaliszyn et al. [14], who observed a 6.7% prevalence of fatty liver in thirty obese girls with PCOS, with age and total testosterone being the primary independent predictors. Subsequently, in a community-based study conducted in Australia, Ayonrinde et al. [15] reported that MASLD was more prevalent in adolescent girls with PCOS compared to those without PCOS (37.5% versus 15.1%, respectively), and that PCOS independently predicted MASLD in this population. These findings have led to the development of a screening tool, termed polycystic ovary syndrome-hepatic steatosis (PCOS-HS) index, which incorporates body mass index (BMI) percentile, waist circumference, alanine aminotransferase (ALT), and sex hormone-binding globulin (SHBG) to assess the risk of MASLD in obese adolescents affected by PCOS [16]. Interestingly, de Zegher et al. [17] suggested that approximately one-third of non-obese adolescents with PCOS treated with oral contraceptives develop MASLD as young women. This finding implies that PCOS in adolescent girls and young women may essentially represent a postmenarcheal central obesity syndrome [17].

In a separate study, Urbano et al. [18] found that 37.5% of adolescent PCOS cases were complicated by MASLD. The authors further demonstrated that individuals with obesity and lower sex-hormone-binding globulin (SHBG) levels were more susceptible to developing hepatic steatosis [16]. In a case–control investigation involving 87 white adolescent girls (47 with PCOS and 40 without), Giannouli et al. [19] reported that MASLD, diagnosed via ultrasound, was more prevalent in adolescents with PCOS compared to controls (22.7% versus 6.1%, respectively). However, no significant differences were observed for hepatic fibrosis, as assessed by transient elastography (TE) [19]. Recent studies have provided more nuanced insights into the MASLD-PCOS relationship.

Accordingly, Kara et al. [20], examining 61 adolescent girls aged 12–18 years diagnosed with PCOS according to the latest guidelines [21,22] and 63 controls with similar age and BMI z-scores who had menstruated regularly for more than two years, found no differences in hepatic steatosis on ultrasound. However, the rate of steatotic liver disease was significantly higher in patients with hyperandrogenic PCOS [20]. Similarly, Patel-Sanchez et al. [23] reported that almost one in five overweight/obese female adolescents (12–18 years old) had MASLD, but PCOS did not increase the risk of this condition. The seemingly contradictory findings in the published literature highlight the complexity of the MASLD-PCOS association and underscore the need for further research. Notably, previous research has primarily focused on the presence of MASLD in PCOS, whereas the occurrence of PCOS in adolescent girls diagnosed with MASLD in pediatric hepatology settings has received less attention.

To address this knowledge gap, we designed a cross-sectional study to investigate the prevalence of PCOS in female adolescents with TE-confirmed MASLD identified in a tertiary care center. Additionally, we examined whether adolescent girls with MASLD and comorbid PCOS differ from those without PCOS in terms of clinical and laboratory parameters. The insights gained from this research have the potential to inform and refine future screening protocols and diagnostic algorithms for these interrelated conditions, specifically tailored to pediatric hepatology settings.

## 2. Materials and Methods

### 2.1. Study Design and Participants

All procedures and visits took place at the Department of Pediatric Gastroenterology and Endocrinology of a tertiary care center (Yuksek Ihtisas Training and Research Hospital, Bursa, Türkiye). This study commenced in February 2021 and concluded in January 2022. Female subjects were considered eligible if they were adolescents aged 12 to 18 years and had a diagnosis of MASLD based on TE findings. Pediatric patients with viral hepatitis B, C, or A, Wilson’s disease, hemochromatosis, and significant systemic illnesses were excluded, as were those using medications known to induce hepatic steatosis. The research protocol adhered to the principles of the Declaration of Helsinki and received approval from the Ethics Committee of KTO Karatay University Faculty of Medicine (approval date and number: 13 July 2021—11873). Prior to participation, written informed consent was obtained from all participants and their parents or legal guardians.

### 2.2. Transient Elastography

In all participants, MASLD was diagnosed using TE performed with the FibroScan^®^ 502 Touch device (Echosens, Paris, France). An experienced pediatric hepatologist, who was blinded to the clinical and laboratory findings, conducted the assessments. The controlled attenuation parameter (CAP) was measured to assess hepatic steatosis, and a CAP value exceeding 225 dB/m was determined as the diagnostic criterion for MASLD [24]. Measurements began with a standard M probe, with automatic probe selection (XL or M) based on the distance between the liver capsule and the skin. Reliability criteria included obtaining at least ten valid measurements and maintaining an interquartile-range-to-median ratio of ≤0.37 [24].

### 2.3. PCOS Screening

All adolescents diagnosed with MASLD were subsequently screened for PCOS in accordance with recent guidelines for diagnosing PCOS in adolescents [21,22]. The diagnosis was established based on the presence of biochemically confirmed hyperandrogenism and irregular menstrual cycles, following the exclusion of other potential causes such as thyroid dysfunction, non-classical congenital adrenal hyperplasia, prolactinoma, androgen-secreting tumors, and pregnancy.

### 2.4. Clinical Characteristics

Weight was measured in kilograms using a calibrated digital scale, with participants wearing light clothing and no shoes. Height was measured in centimeters using a wall-mounted stadiometer. BMI was calculated as weight (kg) divided by height squared (m^2^). Waist circumference (WC) was measured in centimeters at the midpoint between the lower margin of the last palpable rib and the top of the iliac crest using a flexible, non-stretching tape. Blood pressure measurements, including systolic (SBP) and diastolic blood pressure (DBP), were taken in mmHg using a calibrated automated device after participants rested for 5 min in a seated position. Two readings were taken 5 min apart, and the average was recorded. Standard deviation scores (SDSs) for weight, height, BMI, SBP, and DBP were calculated for all participants. Obesity status was determined based on BMI percentiles for age and sex, categorized as normal, overweight, or obese according to standard pediatric definitions. Metabolic syndrome was diagnosed using a modified version of the NCEP ATP III criteria tailored for adolescents, which require the presence of at least three of the following conditions: central obesity (waist circumference at or above the 90th percentile for age and sex), elevated triglycerides (≥150 mg/dL), low HDL cholesterol (≤40 mg/dL), hypertension (blood pressure at or above the 90th percentile for age, sex, and height), and impaired fasting glucose (≥100 mg/dL). Clinical evaluations also included diagnosing metabolic syndrome and conducting a visual examination for acanthosis nigricans, performed by trained clinicians.

### 2.5. Laboratory Parameters

Blood samples were obtained through venipuncture following an overnight fast. A comprehensive panel of parameters was analyzed using standard laboratory techniques, encompassing a complete blood count (including white blood cells, hemoglobin, platelets, and mean platelet volume), metabolic markers (fasting blood glucose and insulin), liver function tests (aspartate aminotransferase [AST], alanine aminotransferase [ALT], gamma-glutamyl transferase [GGT], and alkaline phosphatase [ALP]), and additional biochemical indicators (uric acid, calcium, phosphorus, and vitamin D). All participants underwent a full lipid profile assessment, which included total cholesterol, triglycerides, low-density lipoprotein cholesterol, and high-density lipoprotein cholesterol. Furthermore, ferritin levels were evaluated to assess iron status, while thyroid function was examined through thyroid-stimulating hormone and free T4 measurements. Insulin resistance was quantified using the homeostatic model assessment of insulin resistance (HOMA-IR), calculated from fasting glucose and insulin levels. Serum concentrations of total and free testosterone, 17-hydroxyprogesterone, dehydroepiandrosterone, and androstenedione were quantified solely in subjects presenting with comorbid PCOS.

Serum fasting glucose, total cholesterol, triglycerides, HDL-cholesterol, LDL-cholesterol, and insulin levels were measured after a 12 h fasting period using the Cobas 8000 analyzer (Roche Diagnostics, Indianapolis, IN, USA). Liver enzymes, including AST, ALT, GGT, and ALP, as well as uric acid levels, were determined through spectrophotometry using the Abbott ARCHITECT platform (Abbott Diagnostics, Abbott Park, IL, USA). Serum ferritin was assessed via chemiluminescent microparticle immunoassay on the same device. Total testosterone and dehydroepiandrosterone sulfate (DHEA-S) were quantified using electrochemiluminescence immunoassays (Elecsys^®^ Testosterone II assay and Elecsys^®^ DHEA-S assay, respectively; Roche Diagnostics GmbH, Mannheim, Germany), whereas androstenedione and 17-hydroxyprogesterone levels were measured by radioimmunoassay (Diagnostic Systems Laboratories, Webster, TX, USA). Novel biomarkers of MASLD—specifically cytokeratin-18 (CK-18) [24] and fibroblast growth factor 21 (FGF-21) [25,26,27]—were measured in duplicate using enzyme-linked immunosorbent assay kits (ELK Biotechnology Lab, Wuhan, China) on serum aliquots stored at −80 °C until analysis.

### 2.6. Screening for Other Causes of Hyperandrogenism and Menstrual Irregularities

Cushing’s syndrome was excluded through a clinical examination and a low-dose dexamethasone suppression test. Androgen-secreting tumors were ruled out based on hormone levels (testosterone > 200 ng/dL, DHEA-S > 800 µg/dL) and imaging studies, including pelvic ultrasound and abdominal CT or MRI as necessary. Additional conditions, such as congenital adrenal hyperplasia, prolactinoma, and thyroid dysfunction, were excluded using specific hormonal assays for 17-hydroxyprogesterone, prolactin, TSH, and free T4.

### 2.7. Statistical Analysis

Categorical variables were summarized using frequencies and percentages, while continuous data were described using descriptive statistics, including the mean, standard deviation, median, minimum, and maximum values. Prior to the application of statistical procedures, the normality of continuous data was assessed using the Kolmogorov–Smirnov test. For comparisons between patients with MASLD with and without comorbid PCOS, Student’s t-test was employed for normally distributed data, whereas the Mann–Whitney U test was used for skewed variables. Categorical variables were compared using the chi-square test. All analyses were conducted using SPSS, version 27.0 (IBM, Armonk, NY, USA), and statistical significance was determined with two-tailed *p*-values < 0.05.

## 3. Results

### 3.1. Clinical Characteristics

This study included 45 female adolescents with TE-confirmed MASLD, of whom 19 (42.2%) had comorbid PCOS and 26 did not (57.8%). Table 1 summarizes the clinical characteristics of the two study groups. Patients with comorbid PCOS exhibited significantly higher weight (108.56 ± 22.63 kg versus 81.45 ± 18.6 kg, *p* = 0.039) and waist circumference (108.56 ± 22.63 cm versus 100.63 ± 19.00 cm, *p* = 0.02) compared to their non-PCOS counterparts. Conversely, patients without comorbid PCOS were significantly taller (161.76 ± 6.69 cm versus 159.50 ± 5.89 cm, *p* = 0.004) and had a higher height-SDS (0.25 ± 1.25 versus 0.10 ± 0.97, *p* = 0.016). Although not statistically significant, patients with comorbid PCOS tended to have a higher BMI (34.02 ± 4.18 kg/m^2^ versus 31.80 ± 6.77 kg/m^2^, *p* = 0.083) and weight-SDS (3.38 ± 1.13 versus 2.84 ± 2.12, *p* = 0.069). No significant differences were observed in the prevalence of obesity (78.9% versus 65.4%, *p* = 0.609) or metabolic syndrome (42.1% versus 34.6%, *p* = 0.609) between the two groups. However, the prevalence of acanthosis nigricans was significantly higher in adolescents with comorbid PCOS (68.4% versus 34.6%, *p* = 0.025).

### 3.2. Laboratory Findings

Laboratory findings of patients with and without comorbid PCOS are presented in Table 2. Hepatic enzymes, including AST (17.00 ± 5.00 U/L versus 19.80 ± 6.16 U/L, *p* = 0.027), ALT (21.00 ± 7.00 U/L versus 23.15 ± 10.27 U/L, *p* = 0.008), and GGT (15.00 ± 5.00 U/L versus 16.98 ± 5.19 U/L, *p* = 0.034), were significantly lower in patients with PCOS. Additionally, serum phosphorus levels were significantly reduced in the PCOS group (3.80 ± 0.50 mg/dL versus 4.56 ± 0.57 mg/dL, *p* = 0.016). However, no significant differences were observed in lipid profiles, glucose metabolism parameters, or novel MASLD biomarkers such as CK-18 and FGF-21 between the two cohorts.

## 4. Discussion

While previous research has extensively examined the prevalence of MASLD in pediatric PCOS cohorts [14,15,16,17,18,19,20], our study uniquely investigates the reverse association, focusing on the occurrence of comorbid PCOS in female adolescents diagnosed with MASLD within a specialized pediatric hepatology setting. The high prevalence of PCOS (42.2%) observed in our MASLD sample is noteworthy. However, this elevated rate is likely attributable to this study’s setting in a tertiary care center, which typically attracts more complex cases. Consequently, the observed prevalence may not be representative of the general pediatric population but rather reflects a highly specific healthcare context. Our clinical findings highlighted intriguing anthropometric differences between MASLD patients with and without comorbid PCOS. As expected, weight and WC were significantly higher in the PCOS group, although the difference in BMI between the groups did not reach statistical significance. The inability of BMI alone to fully capture the presence of comorbid PCOS in female adolescents with MASLD suggests that BMI in isolation is an insufficient biomarker for abdominal adiposity [28] in this population. Given that WC assessment is straightforward to implement clinically [28] and demonstrated a significant association with PCOS, routine inclusion of WC measurement in pediatric hepatology settings may be clinically beneficial for risk stratification and counseling regarding a higher-risk phenotype of pediatric MASLD. However, the observed shorter stature in female adolescents with comorbid NAFLD and PCOS in our study presents an intriguing contrast to previous findings. Notably, Aarestrup et al.’s large-scale study of 65,665 girls from the Copenhagen School Health Records Register reported that persistently tall girls or those transitioning from tall to average height had a higher risk of developing PCOS [29]. This apparent discrepancy may be attributed to several factors. Our focus on a cohort with pediatric MASLD, as opposed to the general population examined by Aarestrup et al. [29], could account for these differences. Additionally, variations in the age ranges studied, potential interactions between MASLD and PCOS, and potential genetic and environmental factors across populations may have contributed to the contrasting results. These considerations highlight the complex interplay between growth patterns, MASLD, and PCOS risk, emphasizing the need for further research to elucidate these relationships across diverse populations and comorbidity profiles. 

Another significant clinical observation in our study is that patients with MASLD and comorbid PCOS had a 4.09-fold higher likelihood of showing acanthosis nigricans—a condition characterized by pigmented, velvety thickening of the skin, particularly in the folds of the groin, armpits, and neck—compared to those without PCOS. This finding emphasizes the importance of thorough dermatological examination in female adolescents with MASLD, as acanthosis nigricans may serve as a clinical marker for comorbid PCOS. Interestingly, despite the increased prevalence of this dermatological condition, no significant differences were observed in HOMA-IR or metabolic syndrome prevalence between the groups with and without PCOS [30]. Although the clinically higher rate of acanthosis nigricans’ development in adolescent MASLD comorbidly complicated with PCOS remains speculative at present, it can be hypothesized that genetic factors may play a role. Specifically, genetic polymorphisms in the fibroblast growth factor receptor 3 gene (*FGFR3*) could be involved. FGFR3 is a gene known to be associated with acanthosis nigricans [31,32], and previous research has suggested that it might also be implicated in the progression of PCOS [33]. Consequently, further exploration of the genetic foundations, with a particular emphasis on *FGFR3* and its interaction with hormonal and inflammatory factors, could shed light on why some adolescents with MASLD and PCOS are more prone to developing acanthosis nigricans.

From a biochemical perspective, the lower liver enzyme levels (AST, ALT, and GGT) observed in patients with comorbid PCOS appear counterintuitive given the metabolic challenges associated with both conditions. However, this observation suggests that the relationship between MASLD and PCOS may involve complex metabolic interactions not fully captured by standard liver function tests. Accordingly, several hypotheses could explain this observation. First, the hyperandrogenic state in PCOS might alter the liver’s response to fat accumulation, potentially leading to a different pattern of enzyme release [34]. Second, in PCOS patients with MASLD, compensatory mechanisms elicited by PCOS may benefit the liver, resulting in the mitigation of hepatocellular dysfunction. For instance, increased concentrations of antioxidants such as α-tocopherol and retinol have been observed in PCOS [35]. These adaptive responses could potentially lead to lower liver enzyme release. Finally, the lower liver enzyme levels observed in this study might reflect an earlier stage of MASLD in patients with comorbid PCOS. Accordingly, it is possible that MASLD in PCOS patients could be at an earlier stage, characterized by hepatic steatosis without significant hepatocellular injury and enzyme release [36].

The lower serum phosphorus levels observed in adolescents with PCOS and MASLD are noteworthy. However, the small sample size of our study limits the ability to draw definitive conclusions about phosphate handling and its potential implications for bone health in this population. While previous research suggests that hormonal imbalances in PCOS, such as elevated androgens, may influence bone metabolism and phosphate regulation [37,38], our limited data warrant cautious interpretation.

Our findings should be approached with caution due to several limitations. The tertiary healthcare setting in which the data were collected may introduce a selection bias towards more complex medical cases. Additionally, the single-center design of this study may limit the generalizability of our results to broader populations. The limited sample size may have restricted our ability to detect subtle intergroup differences, thereby increasing the risk of type II errors. Furthermore, the cross-sectional nature of our investigation precludes the establishment of causal relationships or the elucidation of long-term implications of our observations. The absence of liver biopsy data to corroborate the TE results represents another limitation, as such data could have provided more comprehensive information regarding the histological features of MASLD in the study cohort. Lastly, our study was not specifically designed to compare hormonal analytes between MASLD patients with and without comorbid PCOS. Consequently, measurements of total testosterone, free testosterone, DHEA-S, 17-OHPG, and androstenedione were only conducted for patients with a confirmed diagnosis of PCOS. This decision was driven by clinical purposes rather than research objectives, resulting in a lack of data for MASLD patients without PCOS. The absence of hormonal values for this specific patient subset limits our ability to speculate on the broader significance of these findings within the MASLD population. This caveat highlights the need for future studies specifically designed to explore hormonal differences in MASLD patients with and without PCOS.

## 5. Conclusions

This study revealed distinct clinical and biochemical profiles in female adolescents with MASLD and comorbid PCOS compared to those without this concomitant diagnosis. These findings have the potential to inform and refine future screening protocols for these interrelated conditions in pediatric populations. Importantly, our findings emphasize the necessity of routine PCOS screening in female adolescents with MASLD, especially in those presenting clinical signs such as acanthosis nigricans. Implementing this approach could facilitate earlier diagnosis and treatment, ultimately improving patient outcomes.

## Figures and Tables

**Table 1 jcm-13-05885-t001:** Clinical characteristics of female adolescents with metabolic dysfunction-associated steatotic liver disease stratified according to the presence or absence of comorbid PCOS.

Variable	Absence of Comorbid PCOS (n = 26)		Presence of Comorbid PCOS (n = 19)		*p*
	Mean ± SD	Median (Min–Max)	Mean ± SD	Median (Min–Max)	
Age, years	15.13 ± 1.5	14.85 (12.4–17.7)	15.85 ± 1.37	15.9 (13.7–18)	0.107
Weight, kg	81.45 ± 18.6	82.0 (45–118)	108.56 ± 22.63	108.0 (84–147)	**0.039 ***
Weight-SDS, kg	2.84 ± 2.12	2.70 (0–7)	3.38 ± 1.13	3.40 (2–5)	0.069 *
Height, cm	161.76 ± 6.69	162.0 (149–175)	159.50 ± 5.89	160.0 (149–170)	**0.004**
Height-SDS	0.25 ± 1.25	0.20 (−2 to −3)	0.10 ± 0.97	0.10 (−1 to −2)	**0.016**
BMI, kg/m^2^	31.80 ± 6.77	31.0 (22–45)	34.02 ± 4.18	34.0 (26–45)	0.083
BMI-SDS, kg/m^2^	2.62 ± 1.75	2.50 (0–6)	2.80 ± 1.05	2.80 (1–4)	0.290 *
SBP, mmHg	122.46 ± 12.75	122.0 (100–150)	110.00 ± 15.00	110.0 (90–140)	0.786 *
SBP-SDS, mmHg	1.23 ± 1.13	1.20 (0–4)	0.10 ± 1.31	0.10 (−2 to −3)	0.747
DBP, mmHg	79.99 ± 11.87	80.0 (60–100)	70.00 ± 10.00	70.0 (50–90)	0.727 *
DBP-SDS, mmHg	1.32 ± 0.96	1.30 (0–3)	0.50 ± 0.70	0.50 (0–2)	0.497 *
WC, cm	100.63 ± 19.00	101.0 (75–130)	108.56 ± 22.63	108.0 (85–150)	**0.02**
BMI category, n (%) Normal weight Overweight Obesity	7 (26.9) 2 (7.7) 17 (65.4)		3 (15.8) 1 (5.3) 15 (78.9)		0.609
Metabolic syndrome, n (%)	9 (34.6)		8 (42.1)		0.609
Acanthosis nigricans, n (%)	9 (34.6)		13 (68.4)		**0.025**

Abbreviations: PCOS, polycystic ovary syndrome; SD, standard deviation; SDS, standard deviation score; BMI, body mass index; SBP, systolic blood pressure; DBP, diastolic blood pressure, WC, waist circumference. Significant *p* values (*p* < 0.05) are shown in bold. *p* values marked with an asterisk (*) were calculated using the Mann–Whitney *U* test for non-normally distributed continuous variables; all other *p* values were calculated using Student’s *t*-test for continuous variables or the chi-square test for categorical variables.

**Table 2 jcm-13-05885-t002:** Laboratory characteristics of female adolescents with metabolic dysfunction-associated steatotic liver disease stratified according to the presence or absence of comorbid PCOS.

Variable	Absence of Comorbid PCOS (n = 26)		Presence of Comorbid PCOS (n = 19)		*p*
	Mean ± SD	Median (Min–Max)	Mean ± SD	Median (Min–Max)	
WBC per µL	9696 ± 2393	9700 (5000–15,000)	7500 ± 2500	7500 (5000–10,000)	0.489
Hemoglobin, g/dL	13.48 ± 1.10	13.5 (12–16)	12.00 ± 1.50	12.0 (10–14)	0.052
Platelets, ×10^3^/µL	364 ± 80	365 (150–550)	350 ± 100	350 (250–450)	0.099
MPV, fL	10.04 ± 1.12	10.0 (8–12)	9.50 ± 1.00	9.5 (8–11)	0.125
FBG, mg/dL	91.98 ± 9.39	92.0 (75–110)	85.00 ± 10.00	85.0 (70–100)	0.585
Insulin, μIU/mL	48.04 ± 29.67	48.0 (10–90)	30.00 ± 15.00	30.0 (15–45)	0.730 *
HOMA-IR	11.01 ± 7.17	11.0 (2–25)	5.00 ± 3.00	5.0 (2–10)	0.629 *
AST, U/L	19.80 ± 6.16	20.0 (10–30)	17.00 ± 5.00	17.0 (10–25)	**0.027 ***
ALT, U/L	23.15 ± 10.27	23.0 (10–45)	21.00 ± 7.00	21.0 (15–30)	**0.008 ***
GGT, U/L	16.98 ± 5.19	17.0 (10–25)	15.00 ± 5.00	15.0 (10–20)	**0.034**
ALP, U/L	151.96 ± 65.03	152.0 (80–300)	80.00 ± 30.00	80.0 (50–110)	0.100 *
Uric acid, mg/dL	5.29 ± 1.32	5.3 (3–8)	4.50 ± 1.00	4.5 (3.5–5.5)	0.640
Calcium, mg/dL	9.68 ± 0.47	9.7 (9–10)	9.30 ± 0.20	9.3 (9–9.5)	0.826
Phosphorus, mg/dL	4.56 ± 0.57	4.6 (4–5)	3.80 ± 0.50	3.8 (3.5–4)	0.016
Vitamin D, ng/mL	11.99 ± 4.37	12.0 (5–20)	6.00 ± 2.00	6.0 (4–8)	0.241 *
Total cholesterol, mg/dL	171.33 ± 28.24	171.0 (120–220)	160.00 ± 20.00	160.0 (140–180)	0.088
Triglycerides, mg/dL	151.03 ± 59.96	151.0 (90–210)	170.00 ± 40.00	170.0 (130–210)	0.946
LDL cholesterol, mg/dL	96.36 ± 25.15	96.0 (70–120)	85.00 ± 15.00	85.0 (70–100)	0.080
HDL cholesterol, mg/dL	47.19 ± 12.94	45.5 (28–81)	49.91 ± 11.75	48 (32–80)	0.475
Ferritin, ng/mL	37.08 ± 24.15	32 (3–84)	44.37 ± 26.47	42 (4–111)	0.342
Total testosterone, ng/dL	N/A	N/A	51.52 ± 19.29	52 (20.50–90.30)	N/A
Free testosterone, pg/mL	N/A	N/A	2.63 ±1.25	2.50 (0.49–5.20)	N/A
DHEA-S, µg/dL	N/A	N/A	329.54 ± 111.56	369.5 (93–464)	N/A
17-OHPG, ng/mL	N/A	N/A	0.58 ± 0.49	0.43 (0.07–1.8)	N/A
Androstenedione, ng/dL	N/A	N/A	1.54 ± 0.61	1.5 (0.46–2.61)	N/A
TSH, mIU/L	2.60 ± 1.58	2.30 (1–7.2)	2.98 ± 1.58	2.43 (1.4–9.1)	0.290 *
fT4, ng/dL	1.13 ± 0.14	1.12 (0.82–1.4)	1.08 ± 0.18	1.04 (0.83–1.66)	0.089 *
CK-18, ng/mL	1.08 ± 1.22	0.46 (0.10–5.47)	0.53 ± 0.45	0.39 (0.17–1.57)	0.363 *
FGF-21, pg/mL	15.74 ± 22.68	8.8 (0–95.3)	5.96 ± 4.37	4.35 (2.4–14.1)	0.161 *

Abbreviations: PCOS, polycystic ovary syndrome; SD, standard deviation; WBC, white blood cells; MPV, mean platelet volume; FBG, fasting blood glucose; HOMA-IR, homeostatic model assessment of insulin resistance; AST, aspartate aminotransferase; ALT, alanine aminotransferase; GGT, gamma-glutamyl transferase; ALP, alkaline phosphatase; LDL, low-density lipoprotein; HDL, high-density lipoprotein; DHEA-S, dehydroepiandrosterone sulfate; 17-OHPH, 17 hydroxyprogesterone; TSH, thyroid stimulating hormone; fT4, free T4; CK-18, cytokeratin-18; FGF-21, fibroblast growth factor 21; N/A, not available. Significant *p* values (*p* < 0.05) are shown in bold. *p* values marked with an asterisk (*) were calculated using the Mann–Whitney *U* test for non-normally distributed continuous variables; all other *p* values were calculated using Student’s *t*-test for continuous variables or the chi-square test for categorical variables.

## Data Availability

The data are available upon request with limitations due to anonymity and ethical considerations.

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
