# Peer review of "Impact of Comorbid Polycystic Ovary Syndrome on Clinical and Laboratory Parameters in Female Adolescents with Metabolic Dysfunction-Associated Steatotic Liver Disease: A Cross-Sectional Study"

_jcm, 2024, doi:10.3390/jcm13195885_

Round 1

Reviewer 1 Report

Comments and Suggestions for Authors

In their retrospective cross-sectional cohort study Keskin et al aimed to address a potential knowledge gap in the clinical and biochemical differences between female adolescents with MASLD and comorbid polycystic ovary syndrome (PCOS) (n = 19) compared to those without PCOS (n = 26). Moreover, this study focused primarily on elucidating a probable impact of comorbid PCOS on clinical and laboratory parameters in female adolescents with metabolic dysfunction-associated steatotic liver disease making use of the Rotterdam diagnostic criteria in selecting their study participants who exhibited at least two out of three reported PCOS diagnostic criteria of 1) ovulatory dysfunction (oligo- and/or anovulation), 2) clinical and/or biochemical signs of hyperandrogenism, and 3) polycystic ovarian morphology on ultrasound or elevated serum anti-Müllerian hormone levels. The study reported an intriguing significantly lower serum levels of the liver enzyme AST, ALT, and GGT that were associated with shorter statures and higher body weights of adolescent patients with comorbid PCOS. This report is generally nicely presented and informative. However, certain articles need to be address for this short report.

Main concerns:

1-    Discussion: lines 252- 255: please elaborate on potential hormonal, genetic, and inflammatory/immune factors you believe might be involved in your presently reported significantly clinically higher rate of acanthosis nigricans development in your adolescent’s female population with MASLD comorbidly complicated with PCOS

2-    Discussion: lines 256- 265: please elaborate and provide supportive evidence in favor of your three suggested hypotheses on plausible mechanisms and factors that could explain your presently reported significantly lower levels of the liver enzyme AST, ALT, and GGT in your adolescent’s female population with MASLD comorbidly complicated with PCOS

3-    Discussion: lines 277- 279: please explain what is meant by type II errors mentioned within the context of your current arguments.    

4-    Conclusions: lines 287- 293: please briefly elaborate and explain how your present results may have the potential to inform and refine future screening protocols and diagnostic algorithms for MASLD comorbidly complicated with PCOS in pediatric populations.  

Author Response

In their retrospective cross-sectional cohort study Keskin et al aimed to address a potential knowledge gap in the clinical and biochemical differences between female adolescents with MASLD and comorbid polycystic ovary syndrome (PCOS) (n = 19) compared to those without PCOS (n = 26). Moreover, this study focused primarily on elucidating a probable impact of comorbid PCOS on clinical and laboratory parameters in female adolescents with metabolic dysfunction-associated steatotic liver disease making use of the Rotterdam diagnostic criteria in selecting their study participants who exhibited at least two out of three reported PCOS diagnostic criteria of 1) ovulatory dysfunction (oligo- and/or anovulation), 2) clinical and/or biochemical signs of hyperandrogenism, and 3) polycystic ovarian morphology on ultrasound or elevated serum anti-Müllerian hormone levels. The study reported an intriguing significantly lower serum levels of the liver enzyme AST, ALT, and GGT that were associated with shorter statures and higher body weights of adolescent patients with comorbid PCOS. This report is generally nicely presented and informative. However, certain articles need to be address for this short report.

  1. Discussion: lines 252- 255: Please elaborate on potential hormonal, genetic, and inflammatory/immune factors you believe might be involved in your presently reported significantly clinically higher rate of acanthosis nigricans development in your adolescent’s female population with MASLD comorbidly complicated with PCOS

Response: Thank you for your pertinent comment. While acanthosis nigricans is strongly linked to insulin resistance, we found no significant differences in our sample of adolescents with MASLD according to the presence or absence of comorbid PCOS. Although the clinically higher rate of acanthosis nigricans development in adolescent MASLD patients comorbidly complicated with PCOS remains speculative at present, it can be hypothesized that genetic factors may play a role. Specifically, genetic polymorphisms in the fibroblast growth factor receptor 3 (FGFR3) could be involved. FGFR3 is a gene known to be associated with acanthosis nigricans, and previous research has suggested that it might also be implicated in the progression of PCOS. Therefore, further investigation into the genetic underpinnings, particularly focusing on FGFR3, could provide insights into why some adolescents with MASLD and PCOS may have a higher propensity for developing acanthosis nigricans. We have commented on this point in the revised “Discussion” section.

  1. Discussion: lines 256- 265: please elaborate and provide supportive evidence in favor of your three suggested hypotheses on plausible mechanisms and factors that could explain your presently reported significantly lower levels of the liver enzyme AST, ALT, and GGT in your adolescent’s female population with MASLD comorbidly complicated with PCOS

Response: Thank you for your valuable comment. In response, the text has been revised as follows: “Accordingly, several hypotheses could explain this observation. First, PCOS is characterized by hyperandrogenism, which may influence liver enzyme production or clearance. Androgens have been shown to exert direct effects on hepatic lipid metabolism [34]. These hormonal factors unique to PCOS could potentially modulate liver enzyme levels in the context of MASLD. Second, in PCOS patients with MASLD, compensatory mechanisms elicited by PCOS may benefit the liver, resulting in the mitigation of hepatocellular dysfunction. For instance, increased concentrations of antioxidants such as α-tocopherol and retinol have been observed in PCOS [35]. These adaptive responses could potentially lead to lower liver enzyme release. Finally, the lower liver enzyme levels observed in your study might reflect an earlier stage of MASLD in patients with comorbid PCOS. Accordingly, it is possible that MASLD in PCOS patients could be at an earlier stage, characterized by hepatic steatosis without significant hepatocellular injury and enzyme release [36]”.

  1. Discussion: lines 277- 279: please explain what is meant by type II errors mentioned within the context of your current arguments.    

Response: In order to clarify this point, the text has been revised as follows: “The limited sample size may have restricted our ability to detect subtle intergroup differences, thereby increasing the risk of type II errors”.

  1. Conclusions: lines 287- 293: please briefly elaborate and explain how your present results may have the potential to inform and refine future screening protocols and diagnostic algorithms for MASLD comorbidly complicated with PCOS in pediatric populations.  

Response: In response to your request, we have revised the text as follows: “Importantly, our findings emphasize the necessity of routine PCOS screening in female adolescents with MASLD, especially in those presenting clinical signs such as acanthosis nigricans. Implementing this approach could facilitate earlier diagnosis and treatment, ultimately improving patient outcomes”.

Reviewer 2 Report

Comments and Suggestions for Authors

The authors examined adolescent females with MASLD and compared groups with and without polycystic ovarian syndrome, and report on metabolic phenotype. They found that subjects with PCOS and MASLD had overall worse metabolic dysregulation than subjects with MASLD and no PCOS. 

While studies have been reported examining subjects with PCOS and comparing incidence/severity of MASLD, these authors approached from a GI/Hepatology view point, and this may be of interest for providers from these specialties, that may be less familiar with PCOS and its known metabolic complications. 

There are some major concerns in the study design and reporting that the authors should address, however. 

First and most importantly, authors use the Rotterdam criteria for diagnosis of PCOS in these adolescent subjects. For many years, its has been clear that presentation in adolescence is different and there are well-described criteria for PCOS diagnosis specifically in this population. Specifically, ultrasound is NOT included as part of diagnosis, nor is AMH, as there are no age specific norms for ovarian size, follicle count or AMH in adolescence. Diagnosis thus must include BOTH of the following: biochemically hyperandrogenism (elevated testosterone levels, ideally by LC/mass spectrometry assay) AND irregular menses, after ruling out other conditions (thyroid, non-classical CAH, prolactinoma, androgen-secreting tumor, pregnancy). Please see these references: 1) Ibanez L et al. "An International Consortium Update: Pathophysiology, Diagnosis and Treatment of Polycystic Ovarian syndrome in Adolescence". Horm Res Paediatr (2017) 88 (6): 371–395. 2) Teede HJ et al. "Recommendations from the 2023 International Evidence Based Guidelines for the Assessment and Management of Polycystic Ovarian Syndrome". J Endocr Metab 2023 Sep 18;108(10):2447-2469.

Other specific concerns:

Intro

Line 72: define SHBG as this is the first usage

Add to introduction the known relationship between PCOS and insulin resistance/diabetes, there is a broad literature on this, including studies in adolescents from Cree-Green lab in Colorado and Witchel lab in Pittsburgh, PA.

The first paragraph is too long, would break into 2-3 shorter sections for ease of reading.    Methods How were hormone levels measured (for diagnosis of PCOS, testosterone level, other androgens, AMH) how was Cushing’s ruled out? How ware androgen-secreting tumors ruled out?    Were pelvic ultrasound done for PCOS diagnosis? How was it performed and who read it? What criteria were used (as above, I would recommend NOT using ultrasound criteria but if using, please give. more details)   Results   Line 189: how was metabolic syndrome diagnosed, as there is no clinical definition/criteria in adolescents    Table 2—were testosterone levels checked in both groups? Can those be presented? 

Were the Fibroscan parameters different between PCOS and non-PCOS? (i.e. was MASLD “worse” in PCOS?) 

Discussion: some of the discussion (about bone health, phosphate handling) is speculative, given the very small sample size here. Would focus on the metabolic parameters (and height, which is interesting). 

Author Response

  1. First and most importantly, authors use the Rotterdam criteria for diagnosis of PCOS in these adolescent subjects. For many years, it has been clear that presentation in adolescence is different and there are well-described criteria for PCOS diagnosis specifically in this population. Specifically, ultrasound is NOT included as part of diagnosis, nor is AMH, as there are no age specific norms for ovarian size, follicle count or AMH in adolescence. Diagnosis thus must include BOTH of the following: biochemically hyperandrogenism (elevated testosterone levels, ideally by LC/mass spectrometry assay) AND irregular menses, after ruling out other conditions (thyroid, non-classical CAH, prolactinoma, androgen-secreting tumor, pregnancy). Please see these references: 1) Ibanez L et al. "An International Consortium Update: Pathophysiology, Diagnosis and Treatment of Polycystic Ovarian syndrome in Adolescence". Horm Res Paediatr(2017) 88 (6): 371–395. 2) Teede HJ et al. "Recommendations from the 2023 International Evidence Based Guidelines for the Assessment and Management of Polycystic Ovarian Syndrome". J Endocr Metab2023 Sep 18;108(10):2447-2469.

Response: We appreciate your feedback concerning the application of the Rotterdam criteria. In response to your suggestion, we have revised our diagnostic criteria for PCOS by excluding ultrasound and AMH measurements, in alignment with the guidelines outlined by Ibanez et al. (2017) and Teede et al. (2023). Consequently, we have updated the “Methods” section to reflect these changes and have provided clarification on the methodology used for testosterone measurement.

  1. Intro Line 72: define SHBG as this is the first usage

Response: Corrected as per suggestion.

  1. Add to introduction the known relationship between PCOS and insulin resistance/diabetes, there is a broad literature on this, including studies in adolescents from Cree-Green lab in Colorado and Witchel lab in Pittsburgh, PA.

Response: In order to address this concern, the first paragraph of the “Introduction” section as been revised as follows: “Polycystic ovary syndrome (PCOS) is closely linked to insulin resistance and an increased risk of type 2 diabetes, particularly among adolescents. Insulin resistance in PCOS intensifies hyperandrogenism, creating a cycle of metabolic and reproductive disturbances. Numerous studies, including those by Purwar et al. [1] and Witchel et al. [2], have shown that adolescents with PCOS often exhibit early signs of insulin resistance, increasing their susceptibility to metabolic syndrome and diabetes. These findings high-light the critical need for monitoring glucose metabolism and insulin sensitivity in young females diagnosed with PCOS”.

  1. The first paragraph is too long, would break into 2-3 shorter sections for ease of reading.   

Response: Thank you for your suggestion. In response, we have amended the first paragraph to improve readability.

  1. Methods. How were hormone levels measured (for diagnosis of PCOS, testosterone level, other androgens, AMH) how was Cushing’s ruled out? How ware androgen-secreting tumors ruled out?    Were pelvic ultrasound done for PCOS diagnosis? How was it performed and who read it? What criteria were used (as above, I would recommend NOT using ultrasound criteria but if using, please give. more details).

Response: Regarding the hormonal assays and other laboratory values, we have expanded the “Laboratory Parameters” section to address the reviewer’s concerns. Additionally, we have included a new paragraph titled “Screening for other causes of hyperandrogenism and menstrual irregularities”, where we clarified the methods used to rule out Cushing’s syndrome and androgen-secreting tumors. Furthermore, we have removed all mentions of ultrasound and anti-Müllerian hormone from the study.

  1. Results, Line 189: how was metabolic syndrome diagnosed, as there is no clinical definition/criteria in adolescents   

Response: Metabolic syndrome was diagnosed using a modified version of the NCEP ATP III criteria tailored for adolescents, which requires the presence of at least three of the following conditions: central obesity (waist circumference at or above the 90th percentile for age and sex), elevated triglycerides (≥ 150 mg/dL), low HDL cholesterol (≤ 40 mg/dL), hypertension (blood pressure at or above the 90th percentile for age, sex, and height), and impaired fasting glucose (≥ 100 mg/dL). Clinical evaluations also included diagnosing metabolic syndrome and conducting a visual examination for acanthosis nigricans, performed by trained clinicians. This has been clarified in the revised “Methods” section.

  1. Table 2—were testosterone levels checked in both groups? Can those be presented? 

Response: Total and free testosterone, 17-hydroxyprogesterone, dehydroepiandrosterone, and androstenedione levels were measured only in patients with comorbid PCOS; the results are now presented in the revised Table 2.

  1. Were the Fibroscan parameters different between PCOS and non-PCOS? (i.e. was MASLD “worse” in PCOS?) 

Response: In our study, we assessed transient elastography (TE) parameters, including liver stiffness measurement (LSM) and the controlled attenuation parameter (CAP), to evaluate the severity of MASLD in both PCOS and non-PCOS groups. We found no statistically significant differences in LSM and CAP between the two groups. Despite the presence of MASLD in both populations, our findings indicate that the severity, as measured by LSM and CAP, was comparable between the PCOS and non-PCOS groups. We believe that further research with larger sample sizes may be necessary to detect any subtle differences in TE parameters between these groups.

  1. Discussion: some of the discussion (about bone health, phosphate handling) is speculative, given the very small sample size here. Would focus on the metabolic parameters (and height, which is interesting). 

Response: We appreciate your pertinent comment. As metalobic parameters and height were already extensively mentioned in the “Discussion” section, we addressed the reviewer’s concerning by toning down the comments on bone health and phosphate handling,as follows: “The lower serum phosphorus levels observed in adolescents with PCOS and MASLD are noteworthy. However, the small sample size of our study limits the ability to draw definitive conclusions about phosphate handling and its potential implications for bone health in this population. While previous research suggests that hormonal imbalances in PCOS, such as elevated androgens, may influence bone metabolism and phosphate regulation [37, 38], our limited data warrants cautious interpretation”

Round 2

Reviewer 1 Report

Comments and Suggestions for Authors

Main concerns:

1- The authors added Total testosterone, Free testosterone, DHEA-S, 17-OHPG, and Androstenedione in the revised manuscript but missed the opportunity to discuss the significance of these clinical and lab parameters to their study objectives and conclusions. In fact the authors missed the opportunity to discuss the relevance of these results at all in their current revisions (exceptions are the brief description in lines 175-176). Also, these newly added androgenic lab data from patients without PCOS has no numerical values as shown in the revised Table 2! Was this related to the detection limits of the electrochemiluminescence immunoassays, and the  radioimmunoassay kits used for assaying the levels of these serum androgens? Please elaborate and address this point in your revisions and provide the commercial source of these electrochemiluminescence immunoassays, and the radioimmunoassay kits if available (lines 185-186)

2- An appropriate citation is needed to support the new arguments on lines 303-304.

3- Please replace "your study" in line 307 with "this study" for the sake of linguistic prescriptivism!       

Author Response

  1. The authors added Total testosterone, Free testosterone, DHEA-S, 17-OHPG, and Androstenedione in the revised manuscript but missed the opportunity to discuss the significance of these clinical and lab parameters to their study objectives and conclusions. In fact the authors missed the opportunity to discuss the relevance of these results at all in their current revisions (exceptions are the brief description in lines 175-176). Also, these newly added androgenic lab data from patients without PCOS has no numerical values as shown in the revised Table 2! Was this related to the detection limits of the electrochemiluminescence immunoassays, and the radioimmunoassay kits used for assaying the levels of these serum androgens? Please elaborate and address this point in your revisions and provide the commercial source of these electrochemiluminescence immunoassays, and the radioimmunoassay kits if available (lines 185-186).

Response: We appreciate your insightful feedback. To clarify, the addition of total testosterone, free testosterone, DHEA-S, 17-OHPG, and androstenedione measurements was implemented in response to a Reviewer’s recommendation. However, it is important to note that these hormonal assays were exclusively conducted for clinical purposes in patients with a confirmed PCOS diagnosis. Accordingly, our study design did not encompass a comparative analysis of these analytes between MASLD patients with and without comorbid PCOS. Consequently, hormonal assays were not performed for patients without PCOS. In Table 2, the absence of data for non-PCOS patients is not attributable to the detection limits of the electrochemiluminescence immunoassays or radioimmunoassay kits employed. Rather, it stems from the fact that these hormones were not assessed in MASLD patients lacking specific clinical indications, such as PCOS. In response to the Reviewer’s concerns, we have addressed this methodological limitation comprehensively in the “Discussion” section of our revised manuscript, as follows: “Lastly, our study was not specifically designed to compare hormonal analytes between MASLD patients with and without comorbid PCOS. Consequently, measurements of total testosterone, free testosterone, DHEA-S, 17-OHPG, and androstenedione were only conducted for patients with a confirmed diagnosis of PCOS. This decision was driven by clinical purposes rather than research objectives, resulting in a lack of data for MASLD patients without PCOS. The absence of hormonal values for this specific patient subset limits our ability to speculate on the broader significance of these findings within the MASLD population. This caveat highlights the need for future studies specifically designed to explore hormonal differences in MASLD patients with and without PCOS”. As per your request, we have included the commercial sources of the electrochemiluminescence immunoassays and radioimmunoassay kits used in our study in the “Laboratory Parameters” section to enhance transparency and reproducibility.

  1. An appropriate citation is needed to support the new arguments on lines 303-304.

Response: Thank you for your comment. For enhanced clarity, we have revised the text as follows: “First, the hyperandrogenic state in PCOS might alter the liver’s response to fat accumulation, potentially leading to a different pattern of enzyme release [34]. Second, in PCOS patients with MASLD, compensatory mechanisms elicited by PCOS may benefit the liver, resulting in the mitigation of hepatocellular dysfunction. For instance, increased concentrations of antioxidants such as α-tocopherol and retinol have been observed in PCOS [35]”. We have replaced reference 34 with a more comprehensive review that elucidates the complex role of androgens in modulating hepatocellular function and, consequently, the release of liver enzymes. After careful consideration, we confirm that reference 35 remains appropriate and relevant to our discussion.

  1. Please replace "your study" in line 307 with "this study" for the sake of linguistic prescriptivism! 

Response: We apologize for the oversight. This has now been amended.

Reviewer 2 Report

Comments and Suggestions for Authors

Authors have addressed all comments and paper is suitable for publication

Author Response

Authors have addressed all comments and paper is suitable for publication.

Response: Thank you for your positive comment.